# A machine learning model for early candidemia prediction in the intensive care unit: Clinical application

Qiang Meng[1], Bowang Chen[1], Yingyuan Xu[2], Qiang Zhang[2], Ranran Ding[1], Zhen Ma[1], Zhi Jin[1], Shuhong Gao[1]*, Feng Qu[1]*

1 Jining No. 1 People's Hospital Affiliated to Shandong First Medical University, Jining, Shandong, China, 2 Pulmonary and Critical Care Medicine, Tengzhou Central People's Hospital, Tengzhou City, Shandong Province, People's Republic of China

* rmyyzzeq@163.com (FQ); mengqiang1985@126.com (SG)

**Data Availability Statement:** All relevant data are within the manuscript and its Supporting Information files.

## Abstract

Candidemia often poses a diagnostic challenge due to the lack of specific clinical features, and delayed antifungal therapy can significantly increase mortality rates, particularly in the intensive care unit (ICU). This study aims to develop a machine learning predictive model for early candidemia diagnosis in ICU patients, leveraging their clinical information and findings. We conducted this study with a cohort of 334 patients admitted to the ICU unit at Ji Ning NO.1 people's hospital in China from Jan. 2015 to Dec. 2022. To ensure the model's reliability, we validated this model with an external group consisting of 77 patients from other sources. The candidemia to bacteremia ratio is 1:1. We collected relevant clinical procedures and eighteen key examinations or tests features to support the recursive feature elimination (RFE) algorithm. These features included total bilirubin, age, platelet count, hemoglobin, CVC, lymphocyte, Duration of stay in ICU and so on. To construct the candidemia diagnosis model, we employed random forest (RF) algorithm alongside other machine learning methods and conducted internal and external validation with training and testing sets allocated in a 7:3 ratio. The RF model demonstrated the highest area under the receiver operating characteristic (AUC) with values of 0.87 and 0.83 for internal and external validation, respectively. To evaluate the importance of features in predicting candidemia, Shapley additive explanation (SHAP) values were calculated and results revealed that total bilirubin and age were the most important factors in the prediction model. This advancement in candidemia prediction holds significant promise for early intervention and improved patient outcomes in the ICU setting, where timely diagnosis is of paramount crucial.

## Introduction

Fever is a common symptom among patients admitted to the intensive care unit (ICU), often attributed to infections caused by variable microorganisms. *Candida* spp. is the leading cause of invasive fungal infections, ranking as the fourth most common microorganisms responsible

**Funding:** Key R&D Program of Jining, grant number 2023YXNS151. The funder had no role in study design, data collection and analysis, decision to publish, or preparation of the manuscript.

**Competing interests:** The authors have declared that no competing interests exist.

for bloodstream infections (BSIs) [1]. Both invasive fungal infections and candidemia predominantly affect critically ill patients, with up to one-third episodes of candidemia cases occurring within the ICU [2]. Candidemia usually causes life-threatening conditions, resulting in increased mortality rate, healthcare costs, and prolonged ICU stays [3,4]. However, distinguishing candidemia from bacteremia is challenging due to the lack of specific clinical symptoms and presentations [5,6]. The current gold standard for candidemia diagnosis is a positive blood culture test [7]. However, blood culture tests have low sensitivity and require extended incubation times, potentially delaying antifungal therapy [8]. Therefore, physicians usually need to decide whether to wait for blood culture result which may delay initiation of antifungal therapy and prescribe empirical antifungal drugs when they suspect candidemia infection based on their experience. Retrospective studies have showed that such delays can substantially enhance mortality rates, with percentages rising from 35% to 80% [9,10].Conversely, initiating empirical antifungal treatment too early may induce antifungal resistance which has become a significant global public health threat.

New diagnostic biomarkers such as (1, 3)-β-D-glucan (BGD) and galactomannan (GM), glycan antigens derived from the fungal cell wall, or antibody recognizing mannan, have been introduced in clinical practice [6]. However, these biomarker tests are often costly and require complex infrastructure, limiting hospitals' ability to conduct such around-the-clock tests [11]. So, they may not provide early diagnosis of candidemia in ICU patients, leaving clinicians without timely information when suspecting a candidemia infection [12]. Furthermore, biomarkers could not accurately predict candidemia in many conditions. BGD yields a high percentage of false-positive results due to transfusion of human blood products, hemodialysis, infection by some Gram-positive bacteria, use of certain beta-lactam antibiotics, cellulose dressings or enteral nutrition, and disruptions of gastrointestinal (GI) tract integrity [13,14]. Therefore, the demand for the development of more rapid and accurate approaches to differentiate candidemia from bacterial infections is not only evident but also urgent, such approaches could aid ICU physicians in making informed decision regarding the initiation of antifungal therapy [15,16].

Recently, clinical decision support based on machine learning (ML) has received increasing attention for early prediction of candidemia. ML techniques possess a unique capacity to process vast quantities of data, enabling them to employ advanced mathematical models to grasp intricate patterns within clinical datasets. This capability equips ML to excel in tasks such as pattern detection and differentiation [17]. Previous studies have unveiled the potential of ML in early candidemia diagnosis. Bhavani SV *et al*. [16] trained a gradient boosting machine (GBM) model for early detection of candidemia in high and low risk groups. Their model outperformed the traditional logistic regression model, demonstrating the effectiveness of ML approaches with a higher area under the receiver operating characteristic (AUC). Similarly, Yoo J *et al*. [18] developed the CanDETEC model, utilizing random forest (RF) algorithms for candidemia risk prediction in patients with malignancy, and validates its efficacy through a single-center retrospective stud applied random forest (RF) models for the early prediction of candidemia in cancer patients. Atamna A *et al*. [19] and Gao Y *et al*. [20] constructed a logistic regression (LR) model to pinpoint risk factors of candidemia in internal medicine wards, offering valuable insights for the clinical management of patients with candidemia and bacteremia. Besides, Hu WH *et al*. [21] developed a two-level stacked generalisation model using multiple machines learning algorithms, including Naïve Bayesian (NB), k-Nearest Neighbour (KNN), logistic regression (LR), and random forest (RF) as base classifiers, with a support vector machine (SVM) as the meta-classifier, to predict 14-day mortality in candidemia patients. However, the aforementioned studies primarily focused on general populations of patients outside the ICU setting. Risk factors for *Candida* bloodstream infections in adult patients have

been extensively described and identified, often observed in hospitalized patients. Neverthe-less, critically ill ICU patients often present with distinct risk factors that are less applicable to distinguishing from bacteremia patients. Furthermore, the majority of studies have relied on traditional linear models to construct their predictive models, which offer only moderate predictive accuracy even in the best performing models. The demand for a more accurate and rapid approach to differentiate candidemia from bacteremia in ICU patients has grown, which will guide timely and effective therapeutic interventions in clinical settings. This study focuses on developing an early prediction model for fungal bloodstream infections in ICU patients, employing various machine learning algorithms. Through extensive retrospective analysis and external validation, the study validates the effectiveness and generalizability of the models. Additionally, the research elucidates the contribution of individual features in the model using SHAP values and constructs a predictive nomogram for diagnosing candidemia, providing clinicians with a valuable decision-making tool.

## Materials and methods

### Human ethics

This study adhered to the guidelines set forth in the TRIPOD (Transparent Reporting of a Multivariable Prediction Model for Individual Prognosis or Diagnosis) [22] for the construction of the multivariable prediction model and its subsequent validation. The study protocol received approval from the Human Ethics Review Committee of Ji Ning NO.1 people's Hospital, assigned the reference number KYLL-202308-134. The ethics review board at Ji Ning No.1 people's Hospital granted an exemption from the acquisition of informed consent, given the retrospective nature of the study. All patients' information was handled confidentially during the data collection and manuscript preparation.

### Patient inclusion and exclusion criteria

Clinical data for model development and internal validation were retrospectively collected from the electronic medical record system at Ji Ning No. 1 People's Hospital, spanning from Jan. 2015 to Dec. 2022. The clinical data for this study were accessed from September 2023 to October 2023. Patients in the candidemia group were included if they met the following criteria: (1) aged $\geq$ 18 years; (2) requirement for intensive care; (3) expected duration of stay in ICU >48 hours; and (4) at least one blood peripheral positive for Candida spp. Candidemia group excluded criteria: (1) patients who co-infected with bacteremia within 7 days before and after the detection of candidemia; (2) patients with fungal infections other than those caused by Candida species, or (iv) patients who died within 48 hours. (3) patients who were pregnant or lactating. To ensure a comparable periods of risk exposure for both groups, each control patient had a hospitalization duration similar to the time at risk of cases, defined as the number of days from hospital admission to the occurrence of candidemia. To ensure comparable risk exposure periods between candidemia cases and control patients, we implemented a meticulous matching process. Candidemia cases were retrospectively identified using the microbiology laboratory database, and only the first episode was considered if a patient experienced multiple episodes. Control patients diagnosed with bacteremia were selected in a 1:1 ratio and matched to candidemia cases based on several criteria, including age (±5 years), sex, date of hospital admission, and duration of hospitalization at the time of the first positive blood culture. The hospitalization duration for candidemia cases was defined as the number of days from hospital admission to the occurrence of candidemia. Similarly, for control patients, hospitalization duration was defined as the time from admission to the onset of bacteremia,

ensuring alignment with the candidemia cases' risk period. This matching strategy aims to create a balanced and representative dataset for the analysis [23].

For external validation, clinical data were obtained from Teng Zhou Central People's Hospital, covering the period from Jan. 2020 to Jun. 2023. The external validation data for this study were accessed from January 2024 to February 2024.

## Data collection

We determined the potential risk factors for Candida infection through reviewing the past literature and discussions among the experts in the research team. BDG and GM tests, as new biomarkers, have been rarely used in the past, so we did not include those as risk factors in our study. Data collected at baseline encompassed various characteristics, including the duration of stay in ICU, antibiotic usage within the previous two weeks, presence of solid cancer, diabetes, history of chemotherapy, recent surgical procedures, use of central venous catheter (CVC), receipt of total parenteral nutrition (TPN), immunosuppressant treatments and other clinical laboratory data. Prior to analysis, we applied data processing techniques. Cases with missing data excessing 50% of features were excluded from the dataset. To ensure the quality and completeness of the dataset for subsequent analysis, missing values in the remaining cases were imputed using the mean value of the training set and testing set, respectively.

## Microbiological test

Blood culture was processed using the automated blood culture system (Bio-Merieux SA, Marcy l'etoile, France). Fungal isolates and bacterial isolates were cultured at 35°C for 48–72 hours. Simultaneously, we conducted gram staining and microscopic examination. Strain identification was carried out using the VITEK2 Compact system (Bio-Merieux SA, Marcy l'etoile, France) for both fungal and bacterial isolates.

## Descriptive analysis

Continuous data were presented as mean± standard deviation (SD) or median (interquartile range, IQR), while categorical data were reported as frequencies and percentages. Univariate analyses were conducted, with continues data analyzed using independent t-test or non-parametric tests, and categorical data analyzed using the Chi-square or Fisher's exact test. The descriptive analyses were performed using IBM SPSS Statistics for Windows, Version 25.0. A significance level of $p < 0.05$ was considered statistically significant.

## Feature selection techniques

Feature selection is a crucial process aimed at identifying the most relevant subset of variables within a dataset. This practice serves to reduce the number of features, enhance the model's generalization ability, and mitigate the risk of overfitting [24]. While machine learning algorithms in our study can automatically rank features of importance, they may occasionally introduce a small number of redundant features due to the inherent randomness in feature selection [25]. To address this problem, we employed recursive feature elimination (RFE) on the training set for feature selection. RFE involves the iterative removing of the least important features, allowing us to identify the optimal feature combination that can improve overall generalization performance [26].

## Model development and external validation

We used a range of machine learning algorithms, including extreme gradient boosting (XGB), logistic regression (LR), support vector machines (SVM), recurrent neural network (RNN) and random forest (RF), to establish candidemia prediction models with the selected variables. Extreme gradient boosting (XGBoost) operates by sequentially constructing an ensemble of decision trees, where each tree corrects the errors of its predecessors. This method uses gradient descent to minimize the loss function and enhance model performance [27]. Logistic Regression (LR) is a form of linear regression aimed at establishing the relationship between a binary response variable (indicating the occurrence of an event) and a set of explanatory variables. It achieves this by using a logistic function to model the probability of the binary outcome [28]. Support Vector Machine (SVM) works by constructing an optimal hyperplane that maximizes the margin between different classes. For nonlinear problems, it employs kernel functions to project data into higher-dimensional spaces for effective classification [29]. Recurrent Neural Networks (RNN) simulate the interconnected structure of neurons. Input data passes through multiple layers of nodes (neurons), which apply weighted sums and activation functions to learn complex patterns and make predictions [30]. Random Forest (RF) is an ensemble method that trains multiple regression trees on different random subsets of the data. At each node, a randomly selected subset of predictors is used, which enhances robustness to predictor correlations [31].

The entire dataset was randomly divided into ten groups. In each iteration, seven groups of patients were assigned to the training cohort, while the remaining three groups were assigned to the testing cohort. These models (XGB, LR, SVM, RNN and RF) underwent ten-fold cross-validation on the training set to determine the optimal hyperparameters. Using the optimal hyperparameters for each model, we trained the models on the training set. Subsequently, we applied these optimized machine learning models to the testing cohort and external cohort for internal and external evaluation.

External validation was performed using an independent set of electronic records of healthy individuals obtained from Teng Zhou Central People's Hospital spanning from 2020 to 2023. Through machine learning algorithm of RFE, we have obtained features for constructing an early stage candidemia predict model. Then, we collected data of the selected features from Teng Zhou Central People's Hospital for external validation of candidemia predict model.

## Evaluating performance and interpretation of models

We use receiver operating characteristic curve (ROC) to predict power of all the models for candidemia diagnosis. The area under the ROC curve (AUCROC) served as a metric to assess their capacity to distinguish candidemia from bacteremia. The best predict model was defined as the one with the highest AUC value. Furthermore, we assessed performance using metrics such as accuracy, precision, recall, and F1-score to comprehensively evaluate the predictive model's effectiveness [30]. Score and accuracy were calculated using a threshold of 0.5. Model calibration was visualized through the calibration curve [32]. These curves plot the mean decile prediction probability for patients in each group. Calibration results were evaluated by examining the proximity of the calibration curve to the identity line (y = x), representing the standard calibration. In addition, we conducted a decision curve analysis to evaluate the potential benefits of clinical decision- making based on RF model predictions.

Given that machine learning model are often considered as black-boxes, understanding the impact of each risk variable on predict model can be challenging. To address this problem, we used Shapley additive explanation (SHAP) value [33] to explain the contribution of each feature to specific prediction made by the optimal model. SHAP value was conducted using the

"create_explainer" and "shap" function in "shapper" R package (version 4.3.2). Finally, we used nomograms to estimate probability of individual candidemia infection using the "nomogram "function in the "rms" R package (version 4.3.2)[34]. The nomogram analysis considered multiple variables while offers a more accurate prediction and enhances decision-making for candidemia prediction.

## Results

### Significant differences in patient parameters

In this study, we included a total of 334 patients, comprising 167 patients with candidemia and 167 with bacteremia, to construct a risk prediction model for early candidemia diagnosis in ICU setting. The patient cohort was divided into a training set that consisted of 70% of the patients (234 patients), and a testing set that comprised the remaining 30% (100 patients). Additionally, data from 77 patients, with 38 diagnosed with candidemia and 39 with bacteremia, were recruited from Teng Zhou Central People's Hospital to serve as an external validation set. The procedure for patient's selection is detailed in Fig 1.

Comparisons of the baseline features between the candidemia and bacteremia group were provided in Table 1. As shown in Table 1, age is significantly difference between candidemia and bacteremia patients. There is no statistically significant difference in comorbidities between candidemia and bacteremia patients. The key characteristics of the patients in the training and testing dataset are presented in Table 2. As shown in Table 2, several features, including age, presence of central venous catheter (CVC), platelet count, use of total parenteral nutrition (TPN), and duration of stay in the ICU, exhibited significant differences between patients with candidemia and those with bacteremia patients, both in training and test sets. However, there was no difference in gender, presence of fever, alanine transaminase (ALT),

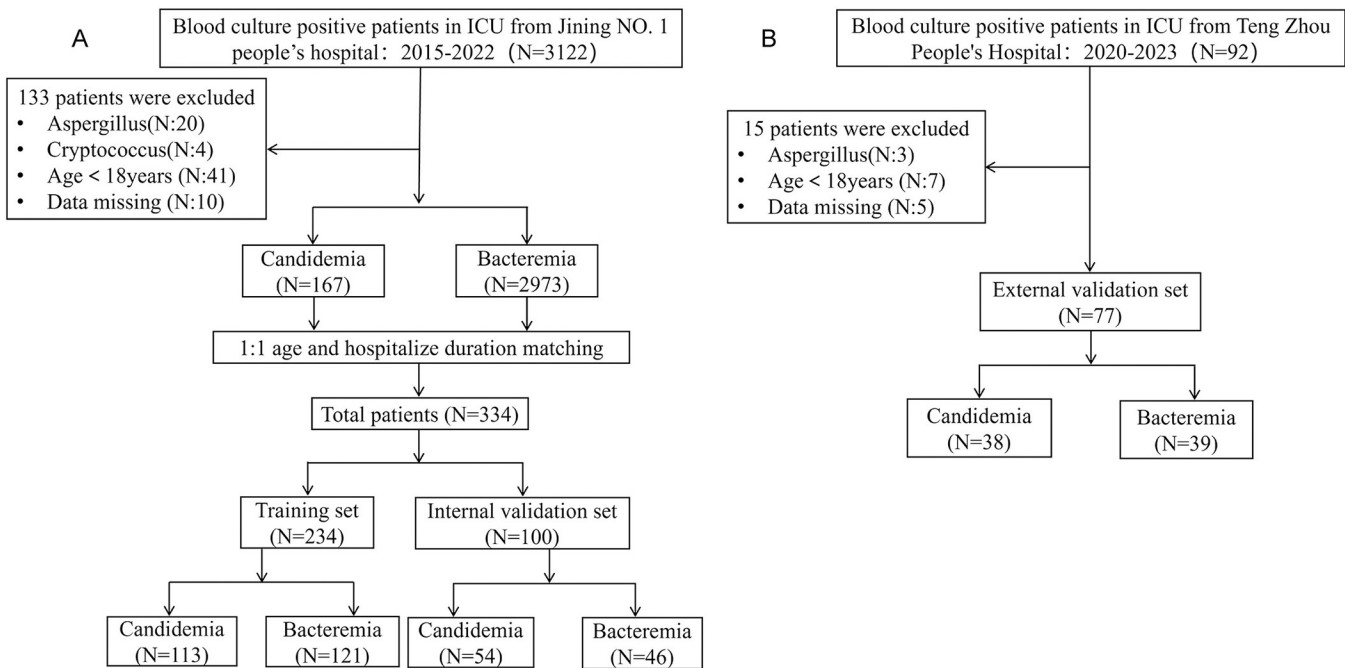

**Fig 1. Flowchart of candidemia and bacteremia patients selected in the ICU for model development and external validation.** (A)Patients select from Jining NO. 1 people's hospital for model development which including the training (n = 234) and internal validation sets (n = 100). (B) Patients select from Teng Zhou People's Hospital for external validation set (n = 77).

**Table 1. Baseline features between the candidemia and bacteremia groups.**

| | Candidemia (n = 167) | Bacteremia (n = 167) | P value |
|---|---|---|---|
| Gender (male, n (%)) | 112(67.6%) | 106(63.7%) | 0.367 |
| Age (y, mean (SD)) | 70.6(17.49) | 64.5(16.85) | 0.036 |
| CHD disease (n (%)) | 48(28.7%) | 46(27.5%) | 0.852 |
| High blood pressure disease (n (%)) | 68(40.7%) | 74(44.3%) | 0.623 |
| Diabetes (n (%)) | 58(34.7%) | 48(28.7%) | 0.239 |
| Chronic pulmonary disease (n (%)) | 48(28.7%) | 46(27.5%) | 0.851 |
| Liver disease (n (%)) | 8(4.79%) | 12(7.14%) | 0.363 |
| Renal disease (n (%)) | 15(45.5%) | 18(54.5%) | 0.582 |
| Cerebrovascular disease (n (%)) | 49(29.3%) | 34(20.3%) | 0.118 |

This table presents the baseline characteristics of patients diagnosed with candidemia (n = 167) and bacteremia (n = 167) in our study. The dataset was randomly split into a training set (70%, n = 234) and a test set (30%, n = 100) to develop and validate the diagnostic models. Statistical comparisons between the two groups were performed using t-tests for continuous variables and chi-square tests for categorical variables. The analysis results showed a significant difference in age between the two groups ($p < 0.05$), while other baseline disease parameters showed no statistical significance. Therefore, these baseline disease parameters were not considered in the model variable selection. Definition of abbreviations: y: Years; CHD: Coronary atherosclerotic heart disease.

aspartate transaminase (AST), albumin, glutamyl transpeptidase (GGT), total bilirubin, creatinine, diabetes, history of solid cancer, chemotherapy, use of immunosuppressive drugs, antibiotic therapy, prior abdominal surgery, white blood count (WBC), neutrophils count, lymphocyte count, C-reactive protein (CRP), procalcitonin (PCT) levels, and hemoglobin level between candidemia and bacteremia patients. A detailed breakdown of the selected features for candidemia and bacteremia in external validation can be found in S1 Table. The comparison of selected features between internal and external validation can be found in S2 Table, WBC and CRP exhibited significant difference.

## Model development and selected features

To identify the most relevant features for our prediction model, the Recursive Feature Elimination (RFE) algorithm, a tree-based method, was utilized to rank the best feature combination on the training set [35]. Finally, the most significant 18 features were selected according to their importance, which include total bilirubin, age, platelet count, hemoglobin, CVC, lymphocyte, duration of stay in ICU, neutrophils, monocyte, antibiotic therapy, PCT, CRP, WBC, immunosuppressive drugs, TPN, abdominal surgery, chemotherapy and solid cancer (Fig 2). These selected features were used for the subsequent model development.

## Model performance for predicting candidemia

We employed multiple prediction models, including Random Forest (RF), Logistic Regression (LR), Support Vector Machine (SVM), extreme gradient boosting (XGB), and Recurrent Neural Network (RNN), to forecast the recurrence of candidemia with the 18 features which were selected. In Table 3, we present a comparative analysis of these models, demonstrating their performance metrics regarding accuracy, precision, recall, F1-score and Area Under the Curve (AUC). The ROC curves were constructed based on the AUC value (Fig 3). For the ML models, we conduct repeated 10-folding cross validation based on the optimization of AUC to identify the optimal model parameters. Our analysis demonstrated the effectiveness of all models, as each exhibited strong predictive performance with an AUC greater than 0.80. Notably, the RF model outperformed the others with an AUC of 0.87. The highest accuracy attained for predicting candidemia was 0.79 achieved by both the RF and RNN models.

**Table 2. Comparison of patient parameters between candidemia and bacteremia groups in the training and the testing sets.**

| | Infection patients(n = 334) | | | | | |
|---|---|---|---|---|---|---|
| | Training set(n = 234) | | *P* value | Testing set (n = 100) | | *P* value |
| | Candidemia (n = 113) | Bacteremia (n = 121) | | Candidemia (n = 54) | Bacteremia (n = 46) | |
| Gender (male, n (%)) | 74(65.4%) | 78(64.5%) | 0.570 | 31(57.4%) | 29(63.0%) | 0.551 |
| Age (y,mean (SD)) | 70.2(16.72) | 64.2(16.31) | 0.027 | 71.7(12.81) | 65.8(15.56) | 0.019 |
| Fever (˚C, mean (SD)) | 38.2(0.68) | 38.6(0.54) | 0.795 | 38.2(0.68) | 38.6(0.56) | 0.124 |
| ALT (U/L, median (IQR)) | 25.9(9.5,58.2) | 23.9(15.1,31.5) | 0.779 | 28.5(12.58,91.78) | 30.95(13.73,60.38) | 0.828 |
| AST (U/L, median (IQR)) | 28.0(17.5, 69.9) | 47.5(18.6, 97.0) | 0.221 | 43.6(27.3, 73.7) | 33.2(21.1, 68.8) | 0.258 |
| Albumin (g/l, mean (SD)) | 28.5(5.60) | 29.4(5.11) | 0.339 | 28.1(6.55) | 31.0(5.92) | 0.068 |
| GGT (U/L, median (IQR)) | 55.0(30.1,107.6) | 49.0(25.5,135.5) | 0.979 | 58.0(37.50,111.5) | 42.95(22.0, 115.5) | 0.079 |
| Total bilirubin(μmol/L, median (IQR)) | 26.8(10.7, 41.7) | 15.5(9.1, 26.9) | 0.037 | 27.3(8.9, 45.8) | 10.8(8.9, 16.1) | 0.028 |
| Creatinine (μmol/L, median (IQR)) | 91.4(68.3, 72.3) | 175.0(43.0,241.8) | 0.046 | 99.0(63.0, 138.0) | 172.5(52.3, 237.0) | 0.035 |
| Diabetes (n, %) | 38(33.6%) | 35(28.9%) | 0.481 | 20(37.0%) | 13(28.3%) | 0.398 |
| Solid cancer (n, %) | 32(28.3%) | 18(14.9%) | 0.012 | 15(27.8%) | 6(13.0%) | 0.071 |
| Chemotherapy (n, %) | 26(23.0%) | 14(11.6%) | 0.094 | 14(25.9%) | 6(13.0%) | 0.126 |
| Immunosuppressive drugs (n, %) | 35(31.0%) | 21(16.4%) | 0.008 | 19(35.2%) | 8(17.4%) | 0.046 |
| Antibiotic therapy (n, %) | 105(92.9%) | 100(82.6%) | 0.017 | 49(90.7%) | 35(76.1%) | 0.046 |
| CVC (n, %) | 68(60.2%) | 31(25.6%) | <0.001 | 31(57.4%) | 10(21.7%) | 0.001 |
| Abdominal surgery (n, %) | 70(61.95%) | 55(45.45%) | 0.012 | 34(63.0%) | 19(41.3%) | 0.044 |
| WBC count (109/L, median (IQR)) | 9.9(6.5,17.3) | 17.4(7.9,20.8) | 0.041 | 11.2(7.6, 20.6) | 19.1(6.5, 24.7) | 0.043 |
| CRP (mg/l, mean (SD)) | 85.3(55.32) | 130.4(80.61) | 0.014 | 91.6(51.57) | 138.6(66.45) | 0.034 |
| PCT (ng/ml, median (IQR)) | 1.9(0.7,10.9) | 3.9(0.3,17.3) | 0.034 | 2.0(0.6, 14.0) | 3.3(0.8, 10.4) | 0.044 |
| Neutrophils count (109/L, median (IQR)) | 7.0(5.6, 14.9) | 10.3(6.1, 14.2) | 0.344 | 7.31(4.90, 10.72) | 10.27(5.75, 17.77) | 0.101 |
| Lymphocyte count (109/L, mean (SD)) | 0.8(0.62) | 1.3(0.73) | 0.036 | 0.67(0.60) | 1.23(0.90) | 0.048 |
| Platelet count (109/L, mean (SD)) | 139.1(109.71) | 186.8(113.31) | 0.011 | 123.6(97.97) | 208.4(119.52) | 0.003 |
| Monocyte count (109/L, median (IQR)) | 0.35(0.24, 0.79) | 0.66(0.36, 0.88) | 0.023 | 0.34(0.19, 0.73) | 0.66(0.43, 1.12) | 0.425 |
| Hemoglobin(g/L, mean (SD)) | 84.1(19.22) | 100.9(21.52) | 0.028 | 79.9(16.41) | 98.9(19.53) | 0.027 |
| TPN (n, %) | 62(54.9%) | 27(22.3%) | <0.001 | 28(51.9%) | 10(21.7%) | 0.004 |
| Duration of ICU stay (days, mean (SD)) | 24.0(21.6) | 14.4(15.5) | 0.046 | 27.5(37.1) | 16.7(35.9) | 0.030 |

This table presents a comparison of key patient parameters, including demographic factors, laboratory values, and clinical characteristics between patients with candidemia and bacteremia in both the training (model development, n = 234) and testing (internal validation, n = 100) datasets. The training set was used for model development, while the testing set was used for internal validation. Statistical analyses were conducted using t-tests for continuous variables and chi-square tests for categorical variables to assess differences between the two groups. Differences with *p*<0.05 were considered statistically significant. This consistency across both sets suggests that the data splitting did not impact the subsequent model development and validation processes. The inclusion of these parameters in the comparison aims to provide a comprehensive overview of the demographic and clinical characteristics of the patients involved in the study. Understanding these characteristics is essential for evaluating the potential impact on the model's performance and its clinical applicability. Definition of abbreviations: y: Years; ALT: Alanine transaminase; AST: Aspartate transaminase; GGT: Glutamyl transpeptidase; TPN: Total parenteral nutrition; CVC: Central venous catheter; WBC: White blood cell; PCT: Procalcitonin; CRP: C-reactive protein; ICU: Intensive care unit; IQR: Interquartile range; SD: Standard deviation.

The calibration curves for our predictive models were presented in Fig 4. The Random Forest and Logistic Regression models showed excellent calibration, with their curves closely aligning with the diagonal line. The mean absolute errors (MAE) were 0.088 and 0.094, respectively. Likewise, the Recurrent Neural Network and Support Vector Machine models exhibit strong calibration performance. However, the calibration curve for the Extreme Gradient Boosting model noticeably deviates from diagonal line, suggesting a potential inaccuracy in comparison to the other models. In addition, we also provide the decision curves that assess the clinical usefulness of the RF model on the test set in Fig 5. These curves show that the RF

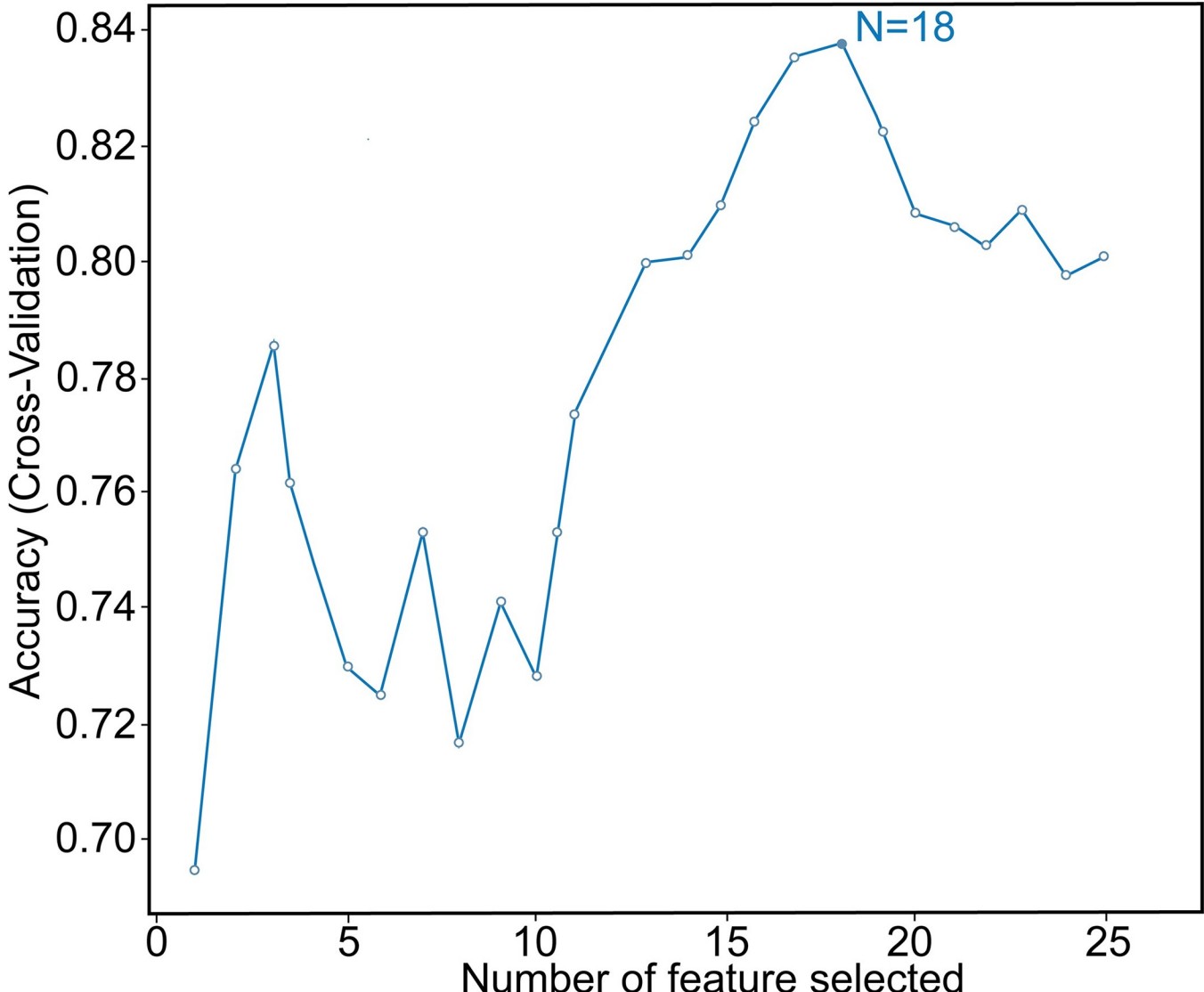

**Fig 2. Recursive feature optimization using the training set.** The dotted line shows the highest accuracy is achieved with 18 features. The horizontal axis is the number of feature selections, and the vertical axis is the prediction accuracy.

**Table 3. Performance of the machine-learning algorithms in internal validation cohort.**

| Model | Accuracy | Precision | Recall | Sensitivity | Specificity | F1 score | AUC |
|---|---|---|---|---|---|---|---|
| Logistic regression | 0.73 | 0.69 | 0.78 | 0.78 | 0.68 | 0.74 | 0.82 |
| Random forest | 0.79 | 0.85 | 0.69 | 0.80 | 0.77 | 0.76 | 0.87 |
| Support vector machine | 0.73 | 0.69 | 0.78 | 0.69 | 0.88 | 0.74 | 0.82 |
| Extreme Gradient boosting | 0.76 | 0.79 | 0.69 | 0.56 | 0.88 | 0.73 | 0.83 |
| Recurrent neural network | 0.79 | 0.78 | 0.78 | 0.78 | 0.79 | 0.78 | 0.84 |

This table summarizes the performance metrics of various machine learning models evaluated using the internal validation cohort (n = 100). Metrics include Accuracy, Precision, Recall, F1 Score, Sensitivity, Specificity and AUC. All models were trained using the same set of variables to ensure comparability. Definition of abbreviations: AUC: Area Under the Curve.

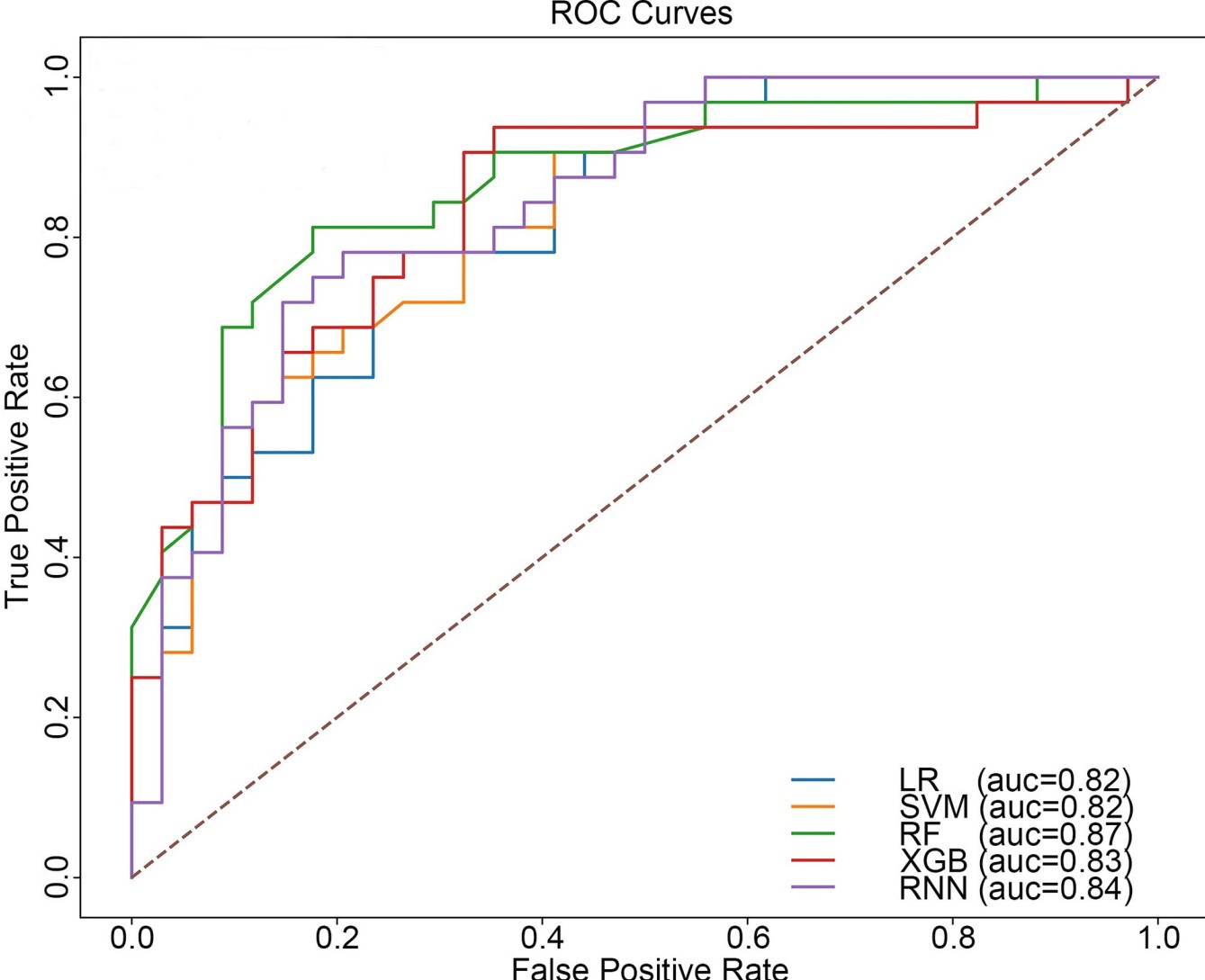

**Fig 3. ROC curves among machine learning models in candidemia diagnostic among ICU patients.** ROC: Receiver operating characteristic curve; auc: Area under the curve; LR: Logistic regression; SVM: Support vector machine; RF: Random forest; XGB: Extreme gradient boosting. RNN: Recurrent neural network.

model offers substantial benefit when the threshold probability of candidemia falls within the range of 55% to 85%.

The Random Forest model continued to perform well again in our external validation cohort from Teng Zhou Central People's Hospital. As shown in Table 4, the RF model achieved a good predictive performance with an AUC of 0.83, slightly lower than the testing set. The RF model exhibited a precision of 0.83 and accuracy of 0.81 in the external validation cohort. These results confirm the model's reliability in correctly discriminating true positives and true negatives.

### Interpretation of random forest predictive model

We used SHAP values to elucidate the contribution of individual features in the Random Forest model. As shown in Fig 6, all features are ranked by their importance. The most influential features in red are positioned at the top, while less significant ones in blue are at the bottom.

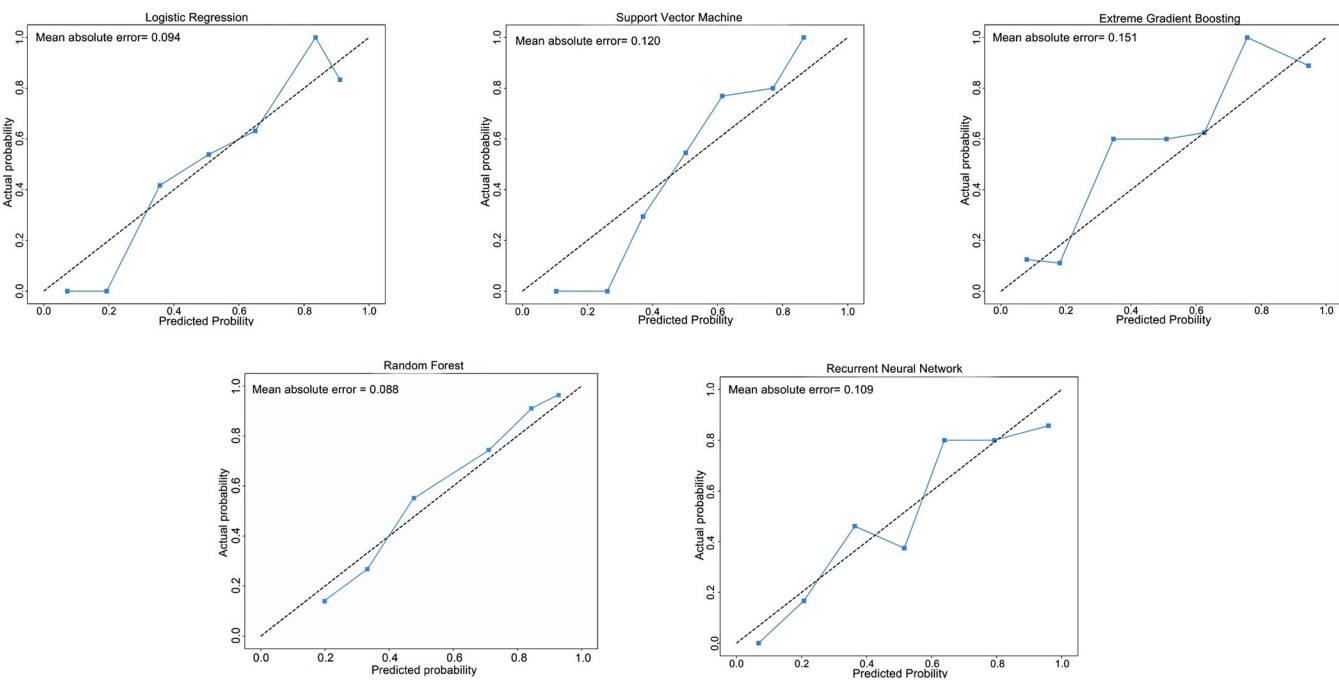

**Fig 4. Calibration curve for model validation of LR, RF, SVM, XGB and RNN on the testing set.** The calibration curve plotted means predicted probabilities for each model against the observed event frequencies on the testing set (n = 100), using decile-binned data.

There are a few key insights from SHAP analysis. First, total bilirubin emerges as the most crucial predictor for candidemia in the RF model, appearing that a higher level of total bilirubin is associated with an elevated risk of candidemia. Second, several other factors positively

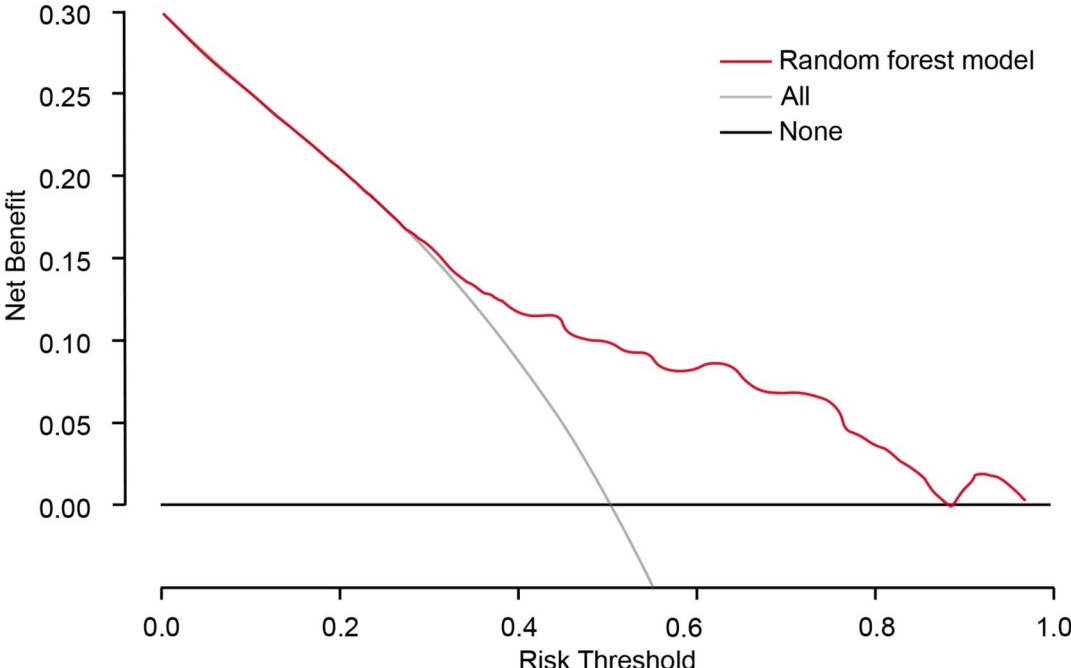

**Fig 5. Decision curve analysis (DCA) of the random forest model on the test set.** X-axis indicates the threshold probability for candidemia diagnosis and Y-axis indicates the net benefit. The red line represents the random forest model.

**Table 4. Performance of machine-learning algorithms in external validation cohort.**

| Model | Accuracy | Precision | Recall | Sensitivity | Specificity | F1 | AUC |
|---|---|---|---|---|---|---|---|
| Logistic regression | 0.73 | 0.63 | 0.63 | 0.74 | 0.65 | 0.63 | 0.80 |
| Random forest | 0.81 | 0.83 | 0.63 | 0.78 | 0.75 | 0.71 | 0.83 |
| Support vector machine | 0.71 | 0.59 | 0.63 | 0.66 | 0.82 | 0.61 | 0.78 |
| Extreme Gradient boosting | 0.80 | 0.68 | 0.81 | 0.59 | 0.82 | 0.74 | 0.83 |
| Recurrent neural network | 0.73 | 0.63 | 0.63 | 0.74 | 0.67 | 0.63 | 0.76 |

This table summarizes the performance metrics of various machine learning models evaluated using the external validation cohort (n = 77). Metrics include Accuracy, Precision, Recall, F1 Score, Sensitivity, Specificity and AUC. All models were trained using the same set of variables to ensure comparability. Definition of abbreviations: AUC: Area Under the Curve.

correlate with the risk of candidemia, including age, antimicrobial therapy, neutrophil count, use of immunosuppressive drugs, total parenteral nutrition, abdominal surgery, chemotherapy, the presence of a central venous catheter, a history of solid cancer, and duration of ICU stay. Third, lower levels of platelets, hemoglobin, lymphocytes, white blood cells, C-reactive protein, procalcitonin, and monocytes are negatively associated with the risk of candidemia.

## The predictive nomogram for candidemia probability

In this study, we constructed a predictive nomogram based on 18 selected features to estimate the probability of candidemia. The primary goal of this analysis is to provide clinicians with a valuable resource for assessing and quantifying the likelihood of candidemia based on specifical clinical characteristics. As shown in Fig 7, each feature is assigned a probability score within a range from 0 to 100. The cumulative scores for all the features are then summed to generate a total score. This total score can then be projected onto the lower risk axis of the nomogram, enabling the prediction of the risk of candidemia diagnosis in ICU patients. To further facilitate clinical use, we plan to convert the content of Fig 5 into a list of point values and include it in the S3 Table.

## Discussion

In this study, we aimed to develop a predicative model for candidemia in intensive care unit patients using machine learning techniques. Our study cohort included 167 patients with candidemia infection and 167 patients with bacteremia infection admitted to the ICU at Ji Ning NO.1 people's hospital from Jan. 2015 to Dec. 2022. Our approach distinguishes itself from prior researches that mainly focused on identifying risk factor for candidemia susceptibility, where healthy people or non-infected patients in medical ward served as the control group. Here, we not only identified risk factors but also constructed a diagnostic model, with bacteremia patients serving as the control model.

When compared our study with Li, et al.'s predictive model for candidemia in cancer patients [36], it is worth noting that while they achieved a slightly higher AUC value, variance in patient selection criteria plays a significant influence. Our study exclusively focused on ICU patients, a group with widely distributed risk factors. By using bacteremia patients within the same ICU as the control group in our model development, we aimed in order to better predicate the candidemia especially in ICU setting. Li et al. also employed bacteremia patients as the control group in their predictive model for invasive candidiasis [37], constituting 81.63% of candidemia and 64.79% of bacteremia cases from ICU, which yielded an AUC value of 0. 92. The slightly lower AUC values in this study may be attributed to the differences in the selected patient populations. An important distinction in our study is the use of data from an external

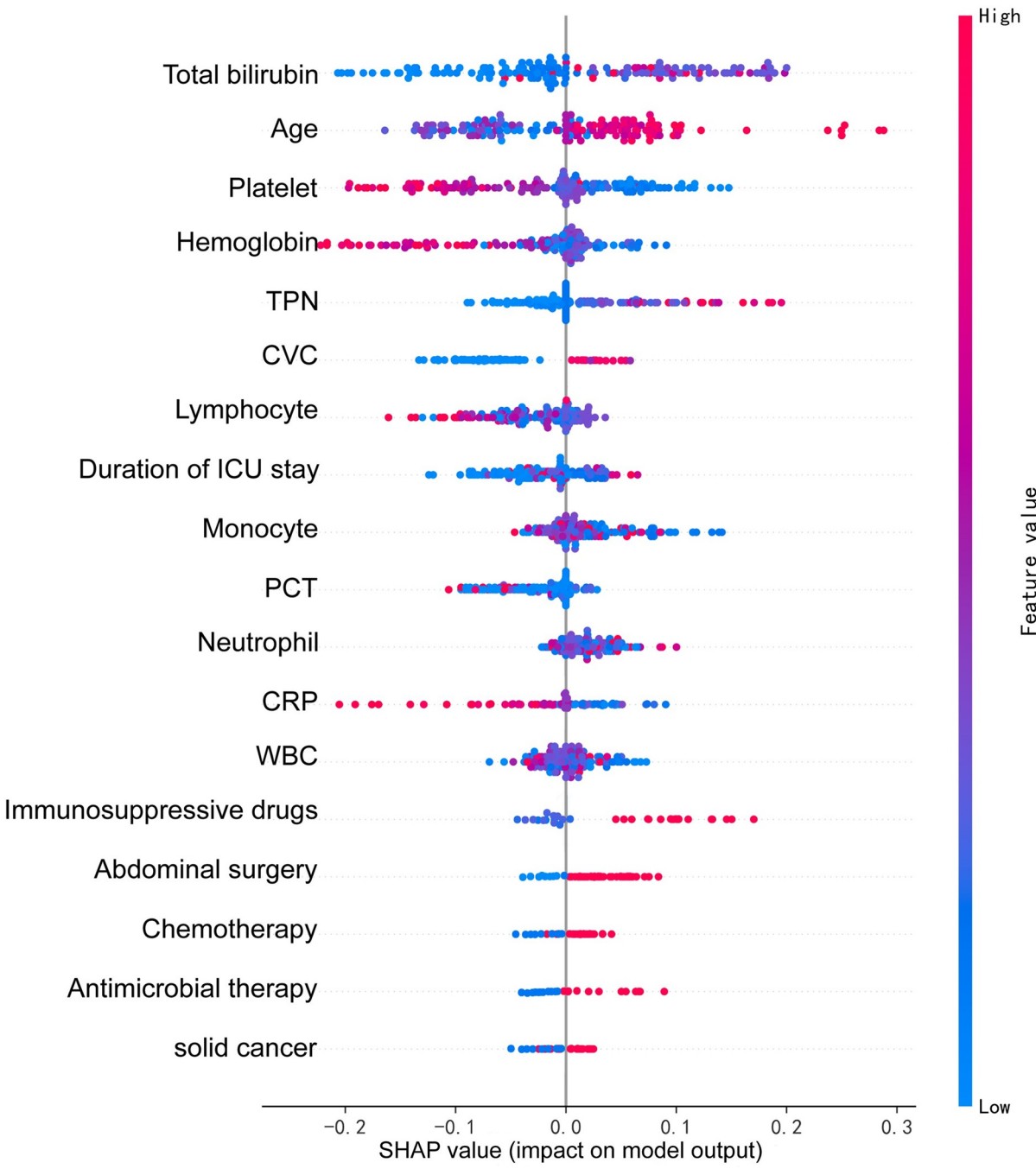

**Fig 6. Feature importance estimated using the Shapley additive explanations (SHAP) values.** The SHAP values were calculated using the training data (n = 234). The plot sorts features by the sum of SHAP value magnitude over all samples. The blue to red color represents the feature value (red high, blue low). The x-axis measures the impacts on the model output (right positive, left negative).

hospital for validation, a practice not commonly observed in previous candidemia prediction study. This external validation provides additional credibility to the generalizability of our model.

We employed a range of machine learning algorithms, including LR, RF, SVM, XGB, and RNN, to construct diagnostic models using routine clinical data. These models displayed

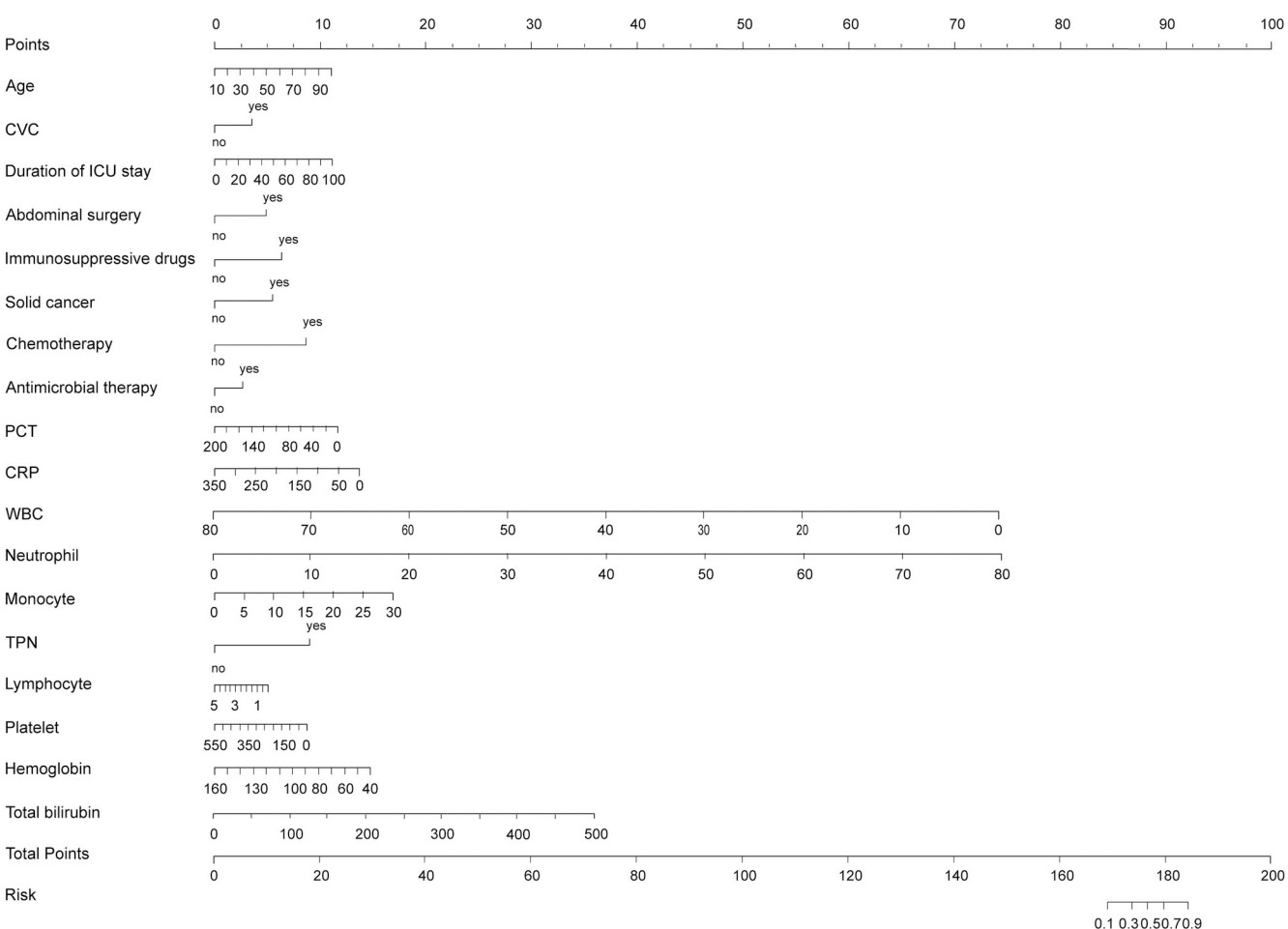

**Fig 7. Nomogram of candidemia predicting using 18 clinical features.** The nomogram was developed using the training data (n = 234). To use the nomogram, we first score each patient's variables to the score axis, added points of all the variables, draw a line from the total score axis to determine candidemia predicting probability on the lower line of the nomogram.

robust performance in distinguishing candidemia from bacteremia among ICU patients. Among these models, the RF model stood out as the most promising candidate. It achieved impressive precision, recall, and F1 scores of 0.85, 0.69, and 0.76, respectively. These metrics were either equal to or greater than the corresponding scores obtained by the RNN model (0.78). When subjected to external validation, the RF model maintained its excellence, yielding an accuracy of 0.81, precision of 0.83, recall of 0.63, and F1 score of 0.71 reflective of its exceptional diagnostic performance. The decision to select the Random Forest (RF) model as the final machine learning model was driven by its remarkable diagnostic performance, highlighted by a high AUC value of 0.87 and these outstanding evaluation metrics. In addition, RF algorithm displays an advantage in handling missing data points, whether they are numerical or factor variables. This capability is particularly valuable in the context of clinical studies, where missing data can be a common occurrence.

The recurrent neural network (RNN) model also showed strong diagnostic performance, with an AUC of 0.84 and the highest accuracy (0.79) among the models tested. RNNs are recognized for their ability to model sequential data and complex temporal patterns, suggesting potential improvements with larger datasets and time-series features. Although not

outperforming the RF model, the RNN's capacity to capture intricate patterns remains a note-worthy advantage, particularly in scenarios involving larger, more dynamic datasets. Notably, prior studies like Cleophas et al. [38] have highlighted the benefits of RNNs in capturing complex relationships, though performance can vary significantly with data quality and quantity.

Logistic regression, traditionally favored for its interpretability, exhibited an AUC of 0.82, which is slightly lower than the more complex models like RF and RNN. This result is consistent with existing literature indicating that logistic regression often provides a baseline level of performance but may not fully capture the complexities of clinical data, especially when interactions between variables are nonlinear and multifaceted. Studies such as Holmes et al. [39] have found logistic regression effective but limited when dealing with complex, high-dimensional data. Nevertheless, its simplicity and ease of interpretation remain beneficial for understanding individual predictors' influence.

Extreme gradient boosting (XGB) achieved an AUC of 0.83, showcasing its ability to handle structured data and select relevant features effectively. While its performance was competitive, the RF model consistently outperformed it across key metrics, highlighting the importance of model selection based on dataset characteristics and clinical needs. Similar studies, such as those conducted by Baxter et al. [40], have demonstrated the strength of XGB in various clinical contexts, although the choice between XGB and RF often depends on specific data characteristics and computational considerations.

The support vector machine (SVM) achieved an AUC of 0.82, similar to logistic regression. Despite SVMs' effectiveness in high-dimensional spaces, their performance was limited, likely due to kernel selection and clinical data complexity. The results are consistent with other studies in the field that report SVM as a powerful classifier, particularly when paired with appropriate kernel functions and parameter tuning [36].

SHAP values were used to identify the crucial predictors of candidemia. Notably, the following factors emerged as positively associated with candidemia risk: high levels of total bilirubin, patient age, total parenteral nutrition (TPN), duration of ICU stay, the presence of a central venous catheter (CVC), and the influence of glucocorticoids. Among these factors, high level of total bilirubin is of most valuable feature. Serum bilirubin is a stable and well-established marker of liver impairment [41,42]. The level of bilirubin is negatively correlated with antimicrobial immunity, rendering individuals more susceptible to candidiasis [43]. Albumin act as an antioxidant and free radical scavenger which can offer protection from inflammatory processes [44]. Patients with hypoalbuminemia are more susceptible to infect opportunistic pathogens like candidemia. However, hypoalbuminemia was not identified as a predict feature for distinguishing between candidemia and bacteremia in this model. In inflammatory states, capillary escape and breakdown of albumin increased which can lead to hypoalbuminemia [44]. Besides, server inflammation often combined with liver failure, which could decrease serum albumin mass because albumin synthesis rate is mainly decreased in liver failure [45]. Low serum albumin levels have been associated with an increased risk of mortality in patients with bloodstream infections because of bacterial or fungal infection [46]. Several studies have demonstrated a correlation between hypoalbuminemia and higher mortality rates in these patients. For example, a study by Lao *et al.* [47] found that low serum albumin levels were independently associated with increased mortality in patients with candidemia. Similarly, a study by A. Capelastegui *et al.* [48] showed that hypoalbuminemia was a significant predictor of mortality in patients with bacteremia.

The remaining positive predicators identified in our study have been supported by previous research in various contexts. For example, both TPN and CVC presence have been recognized as an independent risk factor for candidemia [2]. TPN induced candidemia may arise from alternations in gut microbiota [49], leading to changes in epithelial barrier function and

subsequent intestinal inflammation [50]. The strong association of CVC presence with candidemia, even after adjusting for TPN, underscores its significance for candidemia [19].

Duration of stay in the ICU has also been proposed as an essential indicator of candidemia [51]. This metric is influenced by various factors, including patient demographics, treatment complexity and comorbidities. Moreover, glucocorticoids, which are used in the management of various immune diseases, carry several side effects, such as insulin resistance, increased susceptibility to severe infections, Cushing's syndrome, and peptic ulcers [52]. Prolonged treatment with high-dose systemic glucocorticoids elevates the risk of candidemia [53].

In addition, our study highlighted the association of recent abdominal surgery, chemotherapy, solid cancer and antimicrobial therapy with candidemia. Surgery may compromise the natural body barrier [54], cytotoxic chemotherapy can disrupt the mucosal integrity [55], and antimicrobial therapy may induce alterations in intestinal flora [2]. All these factors collectively create an environment conducive to Candida invasion through mesenteric circulation [2,55].

Platelets are widely recognized as integral components of the host's defense system against invasive infections [56]. In our study, low platelet count was positively associated with candidemia prediction. This finding is significant since platelets paly a protective role in integrating vascular endothelial cells against Candida–induced damage [56]. The decreased presence of platelets may then increase the risk of candidemia. PCT and CRP are well-known for their predictive value in assessing the prognosis of patients with candidiasis and bacterial infections. Some studies have even attempted to use these two markers to distinguish candidemia from bacteremia [57]. Like PCT and CRP, lymphocytes, WBC and monocytes were negatively correlated with candidemia in this study. These findings are particularly intriguing, as white blood play a pivotal role in initiation and maintenance of phagocyte-dependent fungal eliminations [58]. Furthermore, lymphocyte subtyping has even been used as a good marker in the early diagnosis of candidemia [59], which we did not included in this study. Interestingly, our study uncovers an association between low hemoglobin level and high level of bilirubin with an increased incidence of candidemia. This observation might be attributed to hemolytic factors produced by Candida, which can lead to rupture of red blood cells upon binding to the cell membrane [60].

This study has several clinical application prospects. Firstly, the developed machine learning models provide a reliable early prediction method for candidemia in intensive care unit (ICU) patients, which could assist clinicians in making timely decisions regarding the initiation of antifungal therapy. By identifying high-risk patients in advance, hospitals can arrange treatment and monitoring equipment in advance, thereby avoiding additional costs and adverse outcomes resulting from delayed diagnosis. Accurately identifying candidemia patients helps hospitals allocate resources more effectively and can reduce unnecessary resource wastage. Secondly, the identification of key risk factors through SHAP analysis reveals potential mechanisms underlying the development of candidemia, providing important insights for implementing personalized interventions for high-risk patients. Additionally, the predictive nomogram offers a convenient tool for clinicians to estimate the probability of candidemia based on patients' clinical characteristics, aiding in the optimization of individualized treatment plans. Overall, this study provides new insights and approaches for improving patient outcomes, optimizing resource utilization, and enhancing the management strategies for fungal bloodstream infections.

There are several limitations in our study. First, the sample size was constrained by the fact that the data was collected from a single medical center. The relatively small sample size may have implications for the generalizability of our findings. Second, our study design was retrospective in nature. This choice was necessitated by the low incidence of candidemia, which limited our ability to incorporate the latest biomarkers into our predictive model. Third,

although our model demonstrated good performance and achieved acceptable results during external validation, it is essential to acknowledge that retrospective studies inherently carry the potential bias. Therefore, there is a need for multiple-center prospective cohorts to expand our understanding and refine the predicative model.

## Conclusions

In this study, we have demonstrated that the machine learning model on the RF algorithm outperforms other algorithms in predicting candidemia. Our findings emphasize the critical importance of considering total bilirubin levels as a key predictor for candidemia, particularly among ICU patients. The RF model's predictive capabilities could serve as a valuable tool for clinicians in the early diagnosis and treatment of candidemia. The results of this study suggest the potential utility of AI-assisted diagnosis for enhancing the clinical management of candidemia, representing a promising avenue for improving patient care in this context.

## Supporting information

**S1 Table. Characteristics of selected features in the external validation group.** Definition of abbreviations: y: Years; TPN: Total parenteral nutrition; CVC: Central venous catheter; WBC: White blood cell; PCT: Procalcitonin; CRP: C-reactive protein; ICU: Intensive care unit; IQR: Interquartile range; SD: Standard deviation; *The t-test for metric variable if data are normally distributed; **the Chi-square test (big sample size) and Fisher's exact test (small sample size) for categorical variables; ***the Mann–Whitney U-test for metric variables if data are not normally distributed.
(DOCX)

**S2 Table. Comparison of selected features between internal and external validation.** Definition of abbreviations: y: Years; TPN: Total parenteral nutrition; CVC: Central venous catheter; WBC: White blood cell; PCT: Procalcitonin; CRP: C-reactive protein; ICU: Intensive care unit; IQR: Interquartile range; SD: Standard deviation; *The t-test for metric variable if data are normally distributed; **the Chi-square test (big sample size) and Fisher's exact test (small sample size) for categorical variables; ***the Mann–Whitney U-test for metric variables if data are not normally distributed.
(DOCX)

**S3 Table. Nomogram points system for clinical variables.** Definition of abbreviations: TPN: Total parenteral nutrition; CRP: C-reactive protein; WBC: White blood cell; CVC: Central venous catheter.
(DOCX)

## Author Contributions

**Conceptualization:** Bowang Chen.

**Data curation:** Qiang Zhang, Zhi Jin.

**Funding acquisition:** Qiang Meng.

**Software:** Ranran Ding, Zhen Ma.

**Supervision:** Shuhong Gao, Feng Qu.

**Validation:** Yingyuan Xu, Qiang Zhang.

**Visualization:** Qiang Meng.

**Writing – original draft:** Qiang Meng, Zhi Jin.

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
