## [Decision Letter · Decision Letter 0]

28 Jun 2024

PONE-D-24-20146A machine learning model for early candidemia prediction in the intensive care unit: Clinical applicationPLOS ONE

Dear Dr. Meng,

Thank you for submitting your manuscript to PLOS ONE. After careful consideration, we feel that it has merit but does not fully meet PLOS ONE’s publication criteria as it currently stands. Therefore, we invite you to submit a revised version of the manuscript that addresses the points raised during the review process.

We look forward to receiving your revised manuscript.

Kind regards,

Giovanni Giordano

Academic Editor

PLOS ONE

Journal Requirements:

Key R&D Program of Jining, grant number 2023YXNS151. 

Additional Editor Comments:

Dear Authors,

please address the following:

Reviewer 1:

1. The introduction and especially the literature review is not sufficient. Authors need to add more state-of-the-art methods.

2. The model development section is not clear, it should include an explaination of the working principle of all the models utilized in this work.

3. The features used as inputs for the SHAP analysis are not mentioned.

4. The results should be compared with state-of-the-art methods.

Reviewer 2:

Major Comments:

1. The authors carried out 10-fold cross-validation of the algorithms by randomly assigning 7/10 groups for training the models and the remaining 3/10 for internal validation. This is considered standard best practices. However, the results in Table 2 are inconsistent with this premise and suggest that there was a single static pair of training and testing data.

2. Related to the previous point, the results in Figure 2 do not clarify which data set was used. Additionally, if the training or internal validation data were used, then there should be multiple curves for each model corresponding to each fold of the cross-validation. The same lack of clarity also applies to Table 3.

3. The nomogram is a nice addition that can improve the clinical utility of the models here. This is a graphical device, which can lack precision when used. I suggest that the authors also include a table showing the coefficients of the nomogram formula for more precise calculation of risk by clinicians.

4. The authors opted to exclude patients with 50% or more missing features but without providing adequate justification or performing sensitivity analyses to assess the impact of using lower or higher cutoffs.

5. Missing features of the remaining patients were imputed using mean values instead of using more refined approaches such as multiple imputation.

6. On the topic of missing data, please explain how clinicians should deal with missing values when using the nomogram or its formula when attempting to calculate the risk of candidemia infection. This is particularly critical since the model uses 18 features, each having a possibility of being missing, and the overall probability of at least one missing feature is likely moderate to high.

7. The bivariate analysis suggests that many (if not most) features are not marginally associated with infection. I suggest trying to build a more compact model and comparing its performance to the RF model presented. This is likely to have sub-optimal performance relative to the RF model, but if the performance is still comparable (say, AUC>0.8) then this might be more useful to clinician, especially given the likelihood of missing discussed previously.

8. Another reason to consider a more parsimonious model is that the sample size may not allow for adequate power to support 18 variables, and it’s likely that the predictive strength is being driven by only a handful of variables.

9. The data was matched based on hospitalization duration, which was defined as the number of days from hospital admission to the occurrence of candidemia. This is unclear since we do not know the exact time of candidemia occurrence, and especially since testing can take a long time.

Minor Comments

There are many grammatical and organizational errors that need to be revised to improve the readability of the paper. I highlighted these as comments on the manuscript. Moreover, the figure and table captions should be further improved to clearly indicate which sample was used and the sample size.

Reviewers' comments:

Reviewer's Responses to Questions

**Comments to the Author**

1. Is the manuscript technically sound, and do the data support the conclusions?

Reviewer #1: Yes

Reviewer #2: Yes

2. Has the statistical analysis been performed appropriately and rigorously? 

Reviewer #1: Yes

Reviewer #2: Yes

3. Have the authors made all data underlying the findings in their manuscript fully available?

Reviewer #1: Yes

Reviewer #2: Yes

4. Is the manuscript presented in an intelligible fashion and written in standard English?

Reviewer #1: Yes

Reviewer #2: Yes

5. Review Comments to the Author

**Reviewer #1: **The authors have done a good job, the work is interesting, and the results have been presented well however, there are some observations:

1. The introduction and especially the literature review is not sufficient. Authors need to add more state-of-the-art methods.

2. The model development section is not clear, it should include an explaination of the working principle of all the models utilized in this work.

3. The features used as inputs for the SHAP analysis are not mentioned.

4. The results should be compared with state-of-the-art methods.

**Reviewer #2: **This paper presents and contrasts predictive models for early detection of candidemia infection that leverage more readily available clinical information in ICU patients. The results suggest that the machine learning algorithms demonstrated strong predictive strengths against testing and validation data, thus allowing clinicians to identify potential infections in a timely manner.

Major Comments:

1. The authors carried out 10-fold cross-validation of the algorithms by randomly assigning 7/10 groups for training the models and the remaining 3/10 for internal validation. This is considered standard best practices. However, the results in Table 2 are inconsistent with this premise and suggest that there was a single static pair of training and testing data.

2. Related to the previous point, the results in Figure 2 do not clarify which data set was used. Additionally, if the training or internal validation data were used, then there should be multiple curves for each model corresponding to each fold of the cross-validation. The same lack of clarity also applies to Table 3.

3. The nomogram is a nice addition that can improve the clinical utility of the models here. This is a graphical device, which can lack precision when used. I suggest that the authors also include a table showing the coefficients of the nomogram formula for more precise calculation of risk by clinicians.

4. The authors opted to exclude patients with 50% or more missing features but without providing adequate justification or performing sensitivity analyses to assess the impact of using lower or higher cutoffs.

5. Missing features of the remaining patients were imputed using mean values instead of using more refined approaches such as multiple imputation.

6. On the topic of missing data, please explain how clinicians should deal with missing values when using the nomogram or its formula when attempting to calculate the risk of candidemia infection. This is particularly critical since the model uses 18 features, each having a possibility of being missing, and the overall probability of at least one missing feature is likely moderate to high.

7. The bivariate analysis suggests that many (if not most) features are not marginally associated with infection. I suggest trying to build a more compact model and comparing its performance to the RF model presented. This is likely to have sub-optimal performance relative to the RF model, but if the performance is still comparable (say, AUC>0.8) then this might be more useful to clinician, especially given the likelihood of missing discussed previously.

8. Another reason to consider a more parsimonious model is that the sample size may not allow for adequate power to support 18 variables, and it’s likely that the predictive strength is being driven by only a handful of variables.

9. The data was matched based on hospitalization duration, which was defined as the number of days from hospital admission to the occurrence of candidemia. This is unclear since we do not know the exact time of candidemia occurrence, and especially since testing can take a long time.

Minor Comments

There are many grammatical and organizational errors that need to be revised to improve the readability of the paper. I highlighted these as comments on the manuscript. Moreover, the figure and table captions should be further improved to clearly indicate which sample was used and the sample size.

6. PLOS authors have the option to publish the peer review history of their article (what does this mean?). If published, this will include your full peer review and any attached files.

Reviewer #1: No

Reviewer #2: No

---

## [Author Response · Author response to Decision Letter 0]

10 Aug 2024

Repones to editor:

1. We have revised the manuscript according to the journal's formatting requirements.

2. The code generated for this study is shared on GitHub at the following URL: https://github.com/mengqiang1985/candidemia-code.

3. The funder had no role in study design, data collection and analysis, decision to publish, or preparation of the manuscript.

4. we have added and validated the ORCID ID for the corresponding author, Feng Qu, in the Editorial Manager system.

Repones to Reviewer 1:

1. We understand the importance of providing a comprehensive introduction and literature review. In line 72, we have added a review of Naïve Bayesian (NB), k-Nearest Neighbour (KNN), logistic regression (LR), random forest (RF), and support vector machine (SVM) in the introduction.

2. We appreciate your suggestions to enhance the clarity of the model development section. In the revised section of line 148-159, we have provided the following explanations:

Extreme gradient boosting (XGBoost) operates by sequentially constructing an ensemble of decision trees, where each tree corrects the errors of its predecessors. This method uses gradient descent to minimize the loss function and enhance model performance [27]. Logistic Regression (LR) is a form of linear regression aimed at establishing the relationship between a binary response variable (indicating the occurrence of an event) and a set of explanatory variables. It achieves this by using a logistic function to model the probability of the binary outcome [28]. Support Vector Machine (SVM) works by constructing an optimal hyperplane that maximizes the margin between different classes. For nonlinear problems, it employs kernel functions to project data into higher-dimensional spaces for effective classification [29]. Recurrent Neural Networks (RNN) simulate the interconnected structure of neurons. In-put data passes through multiple layers of nodes (neurons), which apply weighted sums and activation functions to learn complex patterns and make predictions [30]. Random Forest (RF) is an ensemble method that trains multiple regression trees on different random subsets of the data. At each node, a randomly selected subset of predictors is used, which enhances robustness to predictor correlations [31].

3. We appreciate your suggestion to specify the input features used in the SHAP analysis. To address this concern, we have updated the manuscript to explicitly list the input features utilized in the SHAP analysis in line 184 of the revised manuscript.

4. We appreciate the reviewer's suggestion to compare our results with state-of-the-art methods. In the revised discussion section, we have revised the discussion section to include detailed comparisons of not only the Random Forest (RF) model but also the Recurrent Neural Network (RNN), Logistic Regression (LR), Extreme Gradient Boosting (XGB), and Support Vector Machine (SVM) models with results from the existing literature:

The recurrent neural network (RNN) model also showed strong diagnostic performance, with an AUC of 0.84 and the highest accuracy (0.79) among the models tested. RNNs are recognized for their ability to model sequential data and complex temporal patterns, suggesting potential improvements with larger datasets and time-series features. Although not outperforming the RF model, the RNN's capacity to capture intricate patterns remains a note-worthy advantage, particularly in scenarios involving larger, more dynamic datasets. Notably, prior studies like Cleophas et al. [38] have highlighted the benefits of RNNs in capturing complex relationships, though performance can vary significantly with data quality and quantity.

 Logistic regression, traditionally favored for its interpretability, exhibited an AUC of 0.82, which is slightly lower than the more complex models like RF and RNN. This result is consistent with existing literature indicating that logistic regression often provides a baseline level of performance but may not fully capture the complexities of clinical data, especially when interactions between variables are nonlinear and multifaceted. Studies such as Holmes et al. [39] have found logistic regression effective but limited when dealing with complex, high-dimensional data. Nevertheless, its simplicity and ease of interpretation remain beneficial for understanding individual predictors' influence.

Extreme gradient boosting (XGB) achieved an AUC of 0.83, showcasing its ability to handle structured data and select relevant features effectively. While its performance was competitive, the RF model consistently outperformed it across key metrics, highlighting the importance of model selection based on dataset characteristics and clinical needs. Similar studies, such as those conducted by Baxter et al. [40], have demonstrated the strength of XGB in various clinical contexts, although the choice between XGB and RF often depends on specific data characteristics and computational considerations.

The support vector machine (SVM) achieved an AUC of 0.82, similar to logistic regression. Despite SVMs' effectiveness in high-dimensional spaces, their performance was limited, likely due to kernel selection and clinical data complexity. The results are consistent with other studies in the field that report SVM as a powerful classifier, particularly when paired with appropriate kernel functions and parameter tuning [36].

Repones to Reviewer 2:

1. Thank you for your observation. We apologize for any confusion caused by our initial description. Here we provide a detailed explanation of our model training and validation process to clarify our methodology:

In our study, we first divided the entire dataset into a training set (70%) and a validation set (30%). We then applied 10-fold cross-validation on the training set to determine the optimal hyperparameters for each model (RF, SVM, RNN, LR, XGB). Therefore, Table 2 only presents a single static pairing of the training and testing data. 

2. We appreciate your feedback and agree that Figure 2 and Table 3 need more clarity. For Figure 2, we utilized the training set data to illustrate the performance of the models. The figure aims to provide a visual representation of model behavior based on the training dataset. We recognize the potential confusion regarding which dataset was used and will ensure this is clarified in the figure caption and accompanying text in the revised manuscript. Regarding Table 3, metrics such as AUC, among others, reflect the model performance after parameter tuning using cross-validation. We employed 10-fold cross-validation to optimize the model parameters, which is standard practice to ensure robust performance assessment. The single result for each metric in Table 3 reflects the aggregate performance across all folds rather than results from each individual fold, providing a clear summary of model performance. 

3. Thank you for your insightful comments regarding the inclusion of a nomogram to improve the clinical utility of our models. We appreciate the suggestion to provide a table with the coefficients of the nomogram formula for precise risk calculation. However, our nomogram is based on a random forest model, which inherently differs from models that provide linear or non-linear formulas, such as logistic regression. Random forest is an ensemble learning method that operates by constructing multiple decision trees during training and outputting the class that is the mode of the classes (classification) or mean prediction (regression) of the individual trees. To further facilitate clinical use, we plan to convert the content of Figure 5 into a list of point values and include it in the S3 Table. This addition will make it easier for clinicians to reference and apply these point values for risk assessment, enhancing the model's practicality. 

4. We appreciate your suggestion to explore the impact of different missing data thresholds. We chose to exclude patients with 50% or more missing features to maintain data integrity and minimize the bias introduced by imputing a large amount of missing data. Retaining samples with higher data completeness ensures the reliability and interpretability of the model's predictions. However, we acknowledge the importance of sensitivity analyses and plan to explore the impact of different thresholds in future studies to assess their influence on model performance.

5. We used mean imputation for missing features to simplify the data processing and reduce computational complexity. However, we recognize that using more refined methods, such as multiple imputation, could improve the accuracy and robustness of the model. Therefore, we plan to apply methods like multiple imputation in future studies to evaluate their impact on model performance.

6. Thank you for highlighting this critical aspect of clinical application. To enhance the accuracy and reliability of the nomogram or its formula in calculating the risk of candidemia infection, we recommend that clinicians ensure the completeness of all 18 features. For the 10 binary variables, clinicians should obtain accurate information based on the patient's actual condition, reference common values from similar patient cohorts, or extract insights from medical records. For other clinical indicators, it is advisable to prioritize obtaining the most recent test data. If real-time data is unavailable, historical data from comparable patients can be used to estimate missing values, and consultation with specialists is encouraged to ensure precision. We acknowledge that incomplete data may occur in clinical practice; thus, clinical judgment and multidisciplinary collaboration are essential for the effective application of the model. We will continue to explore and validate optimal strategies for handling missing data to enhance the model's applicability and reliability across various clinical settings.

7. We appreciate your suggestion to explore a more compact model. We attempted to build a more compact model using variables closely related to infection, such as CRP, WBC, PCT, platelet count, lymphocytes, neutrophils, and monocytes. However, the simplified model achieved an AUC of only 0.645, which is below the clinically useful threshold of AUC > 0.8. This performance gap suggests that the reduced feature set may not capture all relevant interactions present in the Random Forest model. We acknowledge the importance of balancing simplicity and performance and plan to explore additional feature combinations and modeling techniques to achieve this. 

8. Thank you for this insight. We attempted to create a more parsimonious model by selecting key variables related to infection. However, the simplified model did not achieve the desired diagnostic accuracy, with an AUC below the clinical utility threshold. This may be due to the inability to capture complex interactions between variables and the limitation of our sample size. Besides, we have employed 10-fold cross-validation to maximize the utility of our data, ensuring robust model validation and reducing the risk of overfitting. We will continue to explore other feature combinations and modeling approaches to enhance model performance, and consider increasing the sample size to improve predictive capability. 

9. Thank you for your thorough review and valuable feedback on our study. Regarding your concerns about the unclear definition of hospitalization duration used in the data matching process, we have carefully considered your comments and have provided clarification in line 106-116 of the revised manuscript.:

To ensure a comparable periods of risk exposure for both groups, each control patient had a hospitalization duration similar to the time at risk of cases, defined as the number of days from hospital admission to the occurrence of candidemia. To ensure comparable risk exposure periods between candidemia cases and control patients, we implemented a meticulous matching process. Candidemia cases were retrospectively identified using the microbiology laboratory database, and only the first episode was considered if a patient experienced multiple episodes. Control patients diagnosed with bacteremia were selected in a 1:1 ratio and matched to candidemia cases based on several criteria, including age (±5 years), sex, date of hospital admission, and duration of hospitalization at the time of the first positive blood culture. The hospitalization duration for candidemia cases was defined as the number of days from hospital admission to the occurrence of candidemia. Similarly, for control patients, hospitalization duration was defined as the time from admission to the onset of bacteremia, ensuring alignment with the candidemia cases' risk period.

This matching approach was designed to address potential diagnostic delays in detecting candidemia, thereby ensuring that the hospitalization duration accurately reflects the true period of risk for each patient.

Minor Comments

1. Thank you for your feedback on the grammatical and organizational issues in our manuscript. We have carefully reviewed the comments you highlighted in the manuscript and made the necessary revisions to improve the readability of the paper. Specifically, we have corrected grammatical errors, including sentence structure, punctuation, and verb tense consistency. Additionally, we reorganized sections to ensure a smooth flow of information, making it easier for readers to understand our research process and findings. We also simplified complex or ambiguous sentences to enhance clarity and readability, ensuring that our scientific arguments are presented clearly and concisely. 

2. Thank you for your comments regarding the selection of variables in Tables 1 and 2. In Table 1, we present the baseline characteristics of age and chronic diseases for all patients. The results show that, apart from age, there are no statistically significant differences in chronic diseases between the candidemia and bacteremia groups. This suggests that chronic diseases do not significantly affect the incidence of candidemia. Based on these findings, we did not include chronic diseases as screening variables for candidemia in Table 2. We believe this approach provides a clear and focused analysis of factors that are more directly relevant to candidemia incidence. We appreciate your feedback and will ensure that these considerations are clearly communicated in the manuscript.

3. Thank you for your valuable feedback. To address your concerns and provide a more comprehensive evaluation of the model's performance, we have added sensitivity and specificity results to Table 3 and Table 4.

4. We have revised the captions to clearly indicate which sample was used and the sample size, ensuring that readers can easily understand the context and details of each figure and table. The specific modifications are as follows:

Fig 1: Flowchart of candidemia and bacteremia patients selected in the ICU for model development and external validation. (A)Patients select from Jining NO. 1 people’s hospital for model development which including the training and internal validation sets. (B) Patients select from Teng Zhou People's Hospital for external validation set.

Table 1：Baseline features between the candidemia and bacteremia groups.

This table presents the baseline characteristics of patients diagnosed with candidemia (n = 167) and bacteremia (n = 167) in our study. The dataset was randomly split into a training set (70%, n=234) and a test set (30%, n=100) to develop and validate the diagnostic models. Statistical comparisons between the two groups were performed using t-tests for continuous variables and chi-square tests for categorical variables. The analysis results showed a significant difference in age between the two groups (p < 0.05), while other baseline disease parameters showed no statistical significance. Therefore, these baseline disease parameters were not considered in the model variable selection. Definition of abbreviations: y: years; CHD: coronary atherosclerotic heart disease.

Table 2：Comparison of patient parameters between candidemia and bacteremia groups in the training and the testing sets.

This table presents a comparison of key patient parameters, including demographic factors, laboratory values, an

---

## [Editor Report · Decision Letter 1]

19 Aug 2024

A machine learning model for early candidemia prediction in the intensive care unit: Clinical application

PONE-D-24-20146R1

Dear Dr. Qu,

We’re pleased to inform you that your manuscript has been judged scientifically suitable for publication and will be formally accepted for publication once it meets all outstanding technical requirements.

Kind regards,

Giovanni Giordano

Academic Editor

PLOS ONE
---

## [Editor Report · Acceptance letter]

30 Aug 2024

PONE-D-24-20146R1 

PLOS ONE

Dear Dr. Qu, 

I'm pleased to inform you that your manuscript has been deemed suitable for publication in PLOS ONE. Congratulations! Your manuscript is now being handed over to our production team.

Kind regards, 

on behalf of

Dr. Giovanni Giordano 

Academic Editor

PLOS ONE